**Data Availability Statement:** All relevant data are within the manuscript and its Supporting Information files.

**Funding:** The author(s) received no specific funding for this work.

# Accuracy of vertical cup-to-disc ratio discrimination among clinical optometry trainees with different years of clinical experience

**Mohd Izzuddin Hairol**[1,2]◦*, **Yun Rou Lee**[1]◦

**1** Optometry & Vision Science Program, Faculty of Health Sciences, Universiti Kebangsaan Malaysia, Kuala Lumpur, Malaysia, **2** Centre for Community Health Studies (ReaCH), Faculty of Health Sciences, Universiti Kebangsaan Malaysia, Kuala Lumpur, Malaysia

◦ These authors contributed equally to this work.
* Izzuddin.hairol@ukm.edu.my

## Abstract

### Purpose

Accurate evaluation of the cup-to-disc ratio is crucial for optometrists and may be influenced by their clinical experience. The study's objective was to compare the thresholds for discriminating vertical cup-to-disc ratio (VCDR) between years 2, 3, and 4 clinical optometry trainees.

### Methods

One hundred fundus photos with various VCDR sizes were selected from a clinic database. The median VCDR (0.43) photo was assigned as the standard, while the other 99 were assigned as the test photos. The participant's task was to discriminate using a 2-alternate-forced-choice paradigm whether the test photos' VCDR were larger or smaller than the standard VCDR. Data were fit with a Weibull function, and three discrimination thresholds were determined: the point of subjective equality (PSE), the range of VCDR uncertainty, and the ability to judge VCDR that was 0.1 unit larger than the standard photo.

### Results

Year 4 trainees had better VCDR discrimination thresholds. However, the difference between the three participant groups was not statistically different for all measurements (PSE: $F_{(2,27)} = 0.43$, $p = 0.657$; VCDR uncertainty range: $F_{(2,27)} = 0.12$, $p = 0.887$), and thresholds for correctly discriminating VCDR 0.1 larger than the standard photo's VCDR: $F_{(2,27)} = 0.69$, $p = 0.512$).

### Conclusion

Although Year 4 optometry trainees performed slightly better at estimating VCDR than their Year 3 and Year 2 peers, the number of years of clinical experiences did not significantly

**Competing interests:** The authors have declared that no competing interests exist.

affect their VCDR discrimination thresholds when 2-dimensional fundus photos were used as stimuli.

## Introduction

The fundus of the eye can be examined using various methods such as direct ophthalmoscopy, fundus biomicroscopy, fundus photography, and optical coherence tomography [1, 2]. Its evaluation allows the detection of any changes that may indicate the occurrence of a pathology. For instance, the optic nerve head (ONH) evaluation is vital as it may be related to the presence of ocular diseases, such as glaucoma. Specifically for the ONH, features such as the optic disc's size, shape and tilt, the optic cup's shape and depth, cup-to-disc ratio (CDR), and neuroretina rim should be inspected during an eye examination [3].

CDR can be measured horizontally or vertically, giving the horizontal cup-to-disc ratio (HCDR) and vertical cup-to-disc ratio (VCDR), respectively. VCDR is usually used as an indication for glaucoma screening [1, 4–6]. An illustration of the ONH and VCDR is shown in Fig 1. VCDR equal to or larger than 0.8, or a difference in VCDR between eyes by more than 0.2, is pathological [7]. Such fundus findings may indicate glaucoma, characterized by damage to the optic nerve, resulting in irreversible blindness. As 6.6% of blindness among the elderly aged 50 years old and above is contributed by glaucoma [8], accurate VCDR evaluation is essential to aid clinical diagnosis.

The technological development of sophisticated tools has helped in capturing high-quality fundus images. However, the judgement of any changes, normal or pathological, is still dependent on the subjective assessment of the clinician especially when automated estimation is not available. The estimation of VCDR can be subjective and a trainee's ability is typically benchmarked against an examiner who has years of clinical experience [9]. As the accuracy of clinical judgements may be correlated with clinical experience [5, 10], it is hypothesized that trainees with more clinical experience are better at discriminating cup-to-disc ratio than their more junior peers. In addition, knowing how trainees with different clinical experiences judge CDRs would help examiners to devise a fairer way to evaluate their clinical judgement.

Therefore, the study's objective was to determine the thresholds for discriminating VCDR between Year 2, 3, and 4 clinical optometry trainees. The thresholds were then compared between these three groups with a different number of years of clinical experience.

## Methodology

### Study participants

This cross-sectional study was carried out from February to July 2021 in the Optometry Clinic, Universiti Kebangsaan Malaysia (the National University of Malaysia) located in Kuala Lumpur. The study population was clinical optometry trainees in the Optometry and Vision Science Programme. They were in their second, third, and fourth year in the optometry programme. At the time of the data collection, all participants had at least two semesters of theoretical knowledge of the anatomy of the retina and practical experience in fundus examination techniques. All participants had a cumulative grade point average (CGPA) of at least 3.00 out of a maximum of 4.00. It is calculated as the mean of the total grade points for every semester divided by the total number of credits where ≥3.00 CGPA reflected an overall academic performance graded as 'distinction'. Trainees who had resit for any modules or had postponed their studies were excluded.

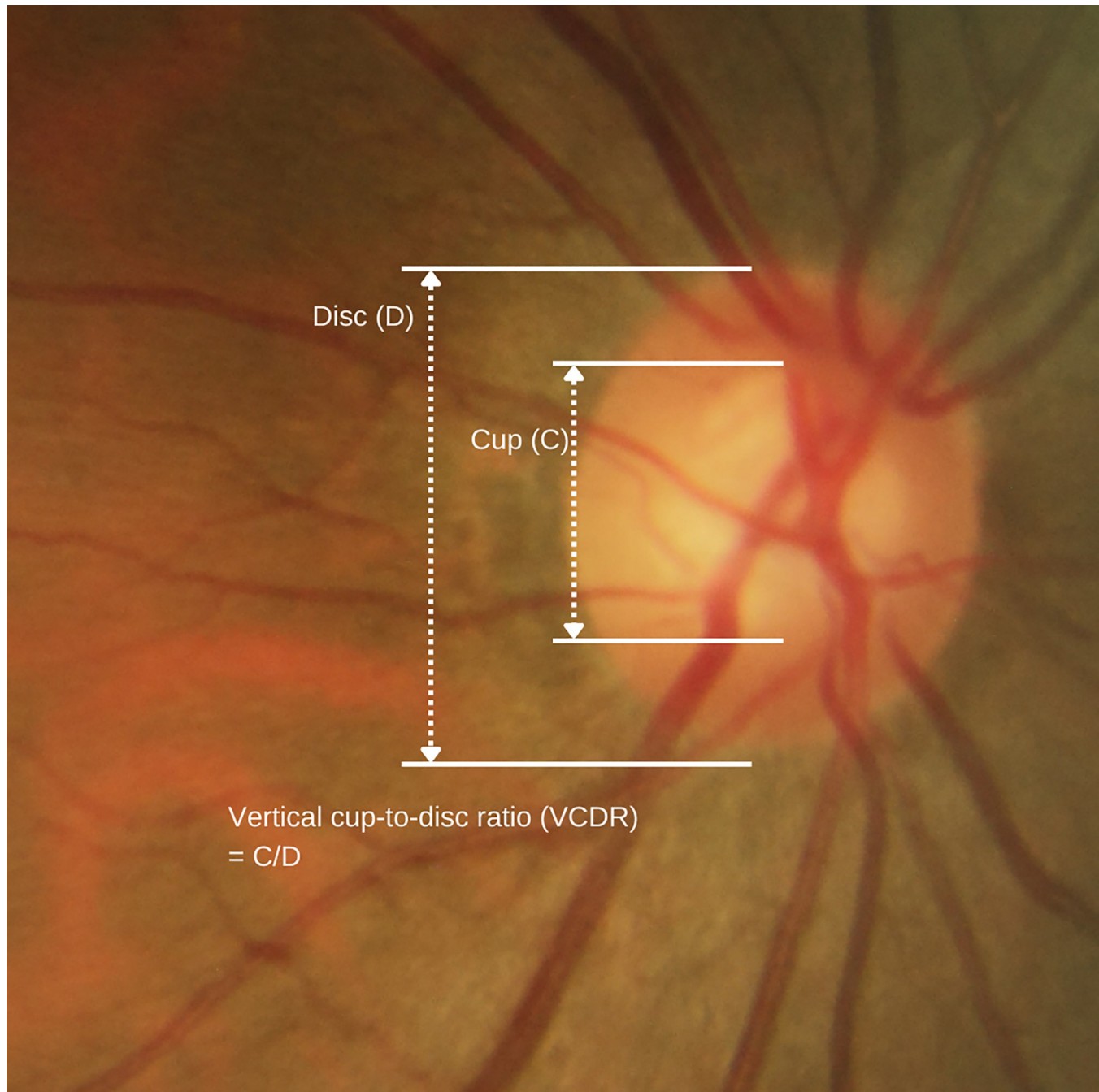

**Fig 1. An illustration of the optic nerve head (ONH) and the vertical cup-to-disc ratio (VCDR).**

## Participant sampling

The population size (N) was 105 trainees. Using G*Power 3.1.9.4 [11], a sample size (n) of 30 would be able to detect a medium to a large effect size of 0.6 with a power of 80% and an error rate α of 0.05. The sample size was divided into three groups based on the participants' study years (Years 2, 3, and 4). Ten trainees who fulfilled the inclusion criteria from each study year were selected using a simple random sampling method. Data were only collected after the

Information Sheet was given to each participant and the Informed Consent Form was signed. The ethics of this study was approved by the institution's Research Ethics Committee (project code: UKM PPI/111/8/JEP-2021-063).

## Experimental stimuli and instrument

All fundus photos were captured with a Canon CR-2 PLUS AF Digital Non-Mydriatic Retinal Camera (Canon U.S.A., Inc.), located in Diagnostic Room 1 of the Optometry Clinic. The fundus photos were first compiled from the instrument's database by author YRL, a final-year optometry student. They were then evaluated and selected by consensus by YRL, author MIH who was a registered optometrist and academician with a 15-year of experience, and another optometrist who was independent of the study. One hundred fundus photos with clear optic disc and optic cup contour, chosen from the Optometry Clinic database (from 2018 to 2020), were selected as stimuli. Fundus photos with any pathology (other than enlarged VCDR) and with apparent artefacts were discarded. Every fundus photo's VCDR was determined using the Canon CR-2 Plus's built-in software.

## Psychophysical procedures

All of the photos were uploaded onto the PsychoPy software [12]. A custom-written program was created to display the stimuli and determine the participants' threshold to discriminate VCDR accurately. The fundus photo whose VCDR was the median (0.43) was chosen as the standard from the 100 photos. The other 99 fundus photos were assigned as the test photos.

For measurements of discrimination thresholds, a test photo was displayed next to the standard photo. Using the 2-alternate-forced-choice (2AFC) method, the participant's task was to indicate if the test photo's VCDR was larger or smaller than the standard photo's VCDR. The discrimination task was carried out for all 99 test photos. The viewing distance was set at 40 cm. At this viewing distance, the visible area of the fundus subtended 14.7 minutes of arc. The procedure was carried out binocularly. Viewing time was unlimited. The participant' responses were collected using a keypress on a keyboard.

All participants underwent a training session to familiarize themselves with the study's procedures. Data from the training session were not included in the analysis. All data collected after the training session were averaged from three experimental runs.

## Determination of VCDR discrimination thresholds

The percent response for "test photo's VCDR judged larger than standard photo's VCDR" was plotted as a function of the test photos' VCDRs in Igor Pro® software. The data was fit with a Weibull function with the formula:

$$f(x) = 100 - 100 \times 2^{-\left(\frac{x}{th}\right)^{\beta}}$$

where $th$ is the estimated VCDR threshold, corresponding to 50% percent response for "test photo's VCDR judged larger than standard photo's VCDR"; $\beta$ is the slope of the psychometric function; and $x$ is a given VCDR.

From the psychometric function, participants' discrimination thresholds were determined in three ways:

i.  The point of subjective equality (PSE), where the test photo's VCDR was judged to be larger than the standard photo's VCDR 50% of the time (Fig 2);

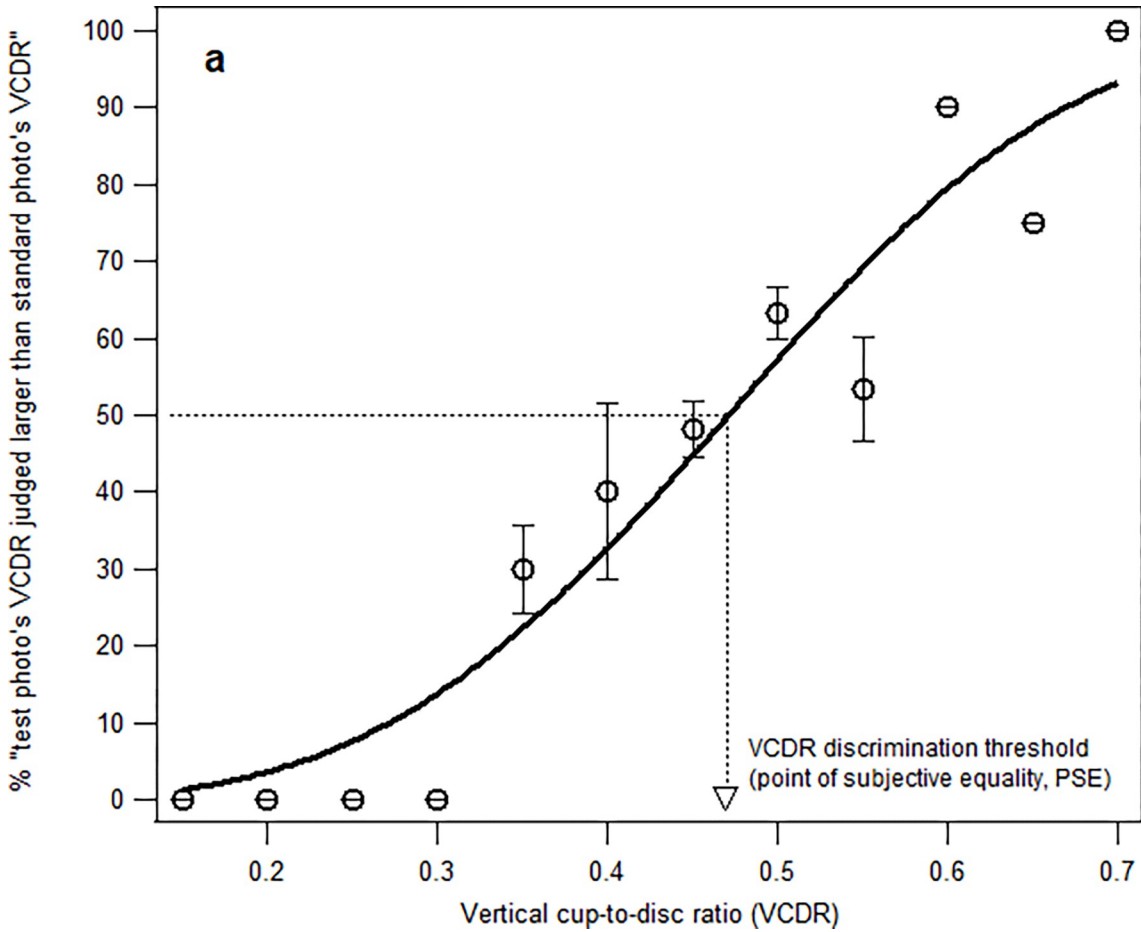

**Fig 2. The percent of test photo's VCDR judged to be larger than standard photo's VCDR, plotted as a function of VCDR.** The data was fit with a Weibull function. The point of subjective equality was determined by finding the VCDR that corresponds to '50% larger' response.

ii. The range of VCDR discrimination uncertainty, which was determined by (a) estimating the test photo's VCDR that was judged to be *larger* than the standard photo's VCDR 75% of the time, and (b) estimating the test photo's VCDR that was judged to be larger than standard photo's VCDR 25% of the time, that is, the VCDR that the participants accurately judged to be 25% *smaller* than the standard photo's VCDR. The range of VCDR discrimination uncertainty was calculated by subtracting (b) from (a) (Fig 3).

iii. The percentage that the participants were able to discriminate that VCDR of test photo was 0.53, that is, 0.1 unit larger than the standard photo's VCDR (0.43) (Fig 4).

## Statistical analysis

Discrimination thresholds for 25%, 50%, and 75% responses, and for correctly discriminating 0.53 VCDR, were calculated for each participant and presented as mean and standard deviation using IBM Statistical Package for Social Sciences version 25 (IBM Corp., Armonk, NY, USA). One-way ANOVA was used to compare the mean discrimination thresholds between three participant groups. The value of significance was set at $p < 0.05$ for all statistical tests.

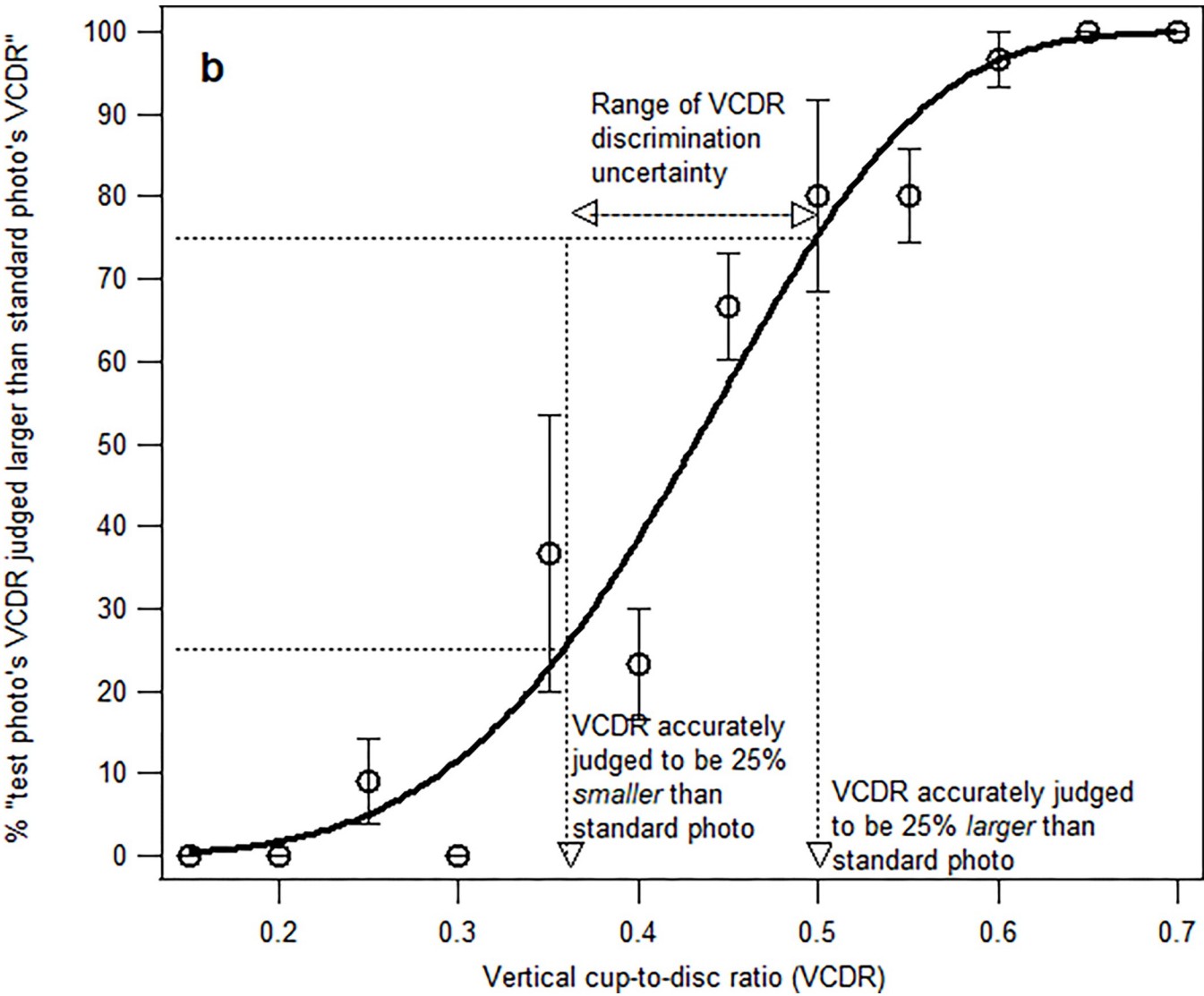

**Fig 3. The range of VCDR discrimination uncertainty was calculated by subtracting the VCDR that was accurately judged to be 25%** *larger* **than the standard photo's VCDR with VCDR that was accurately judged to be 25%** *smaller* **than the standard photo's VCDR.**

## Results

The mean age of the 30 participants was 22.17±0.91 years (age range: 21 to 24 years old). Five participants (16.7%) were males, and twenty-five (85.3%) were females. The characteristics of the participants are summarized in Table 1. There was no significant difference between the year groups' CGPAs (one-way ANOVA [$F_{(2,27)}$ = 2.071, p = 0.146]).

The mean VCDR PSE for participants in Year 2, Year 3, and Year 4 was 0.44±0.04, 0.43±0.06 and 0.43±0.02, respectively. There was no statistically significant difference of PSE between the three groups (one-way ANOVA [$F_{(2,27)}$ = 0.426, p = 0.657]).

The VCDR uncertainty range was determined by subtracting the threshold for correctly discriminating a test photo's VCDR that was 25% larger than the standard photo's VCDR with the threshold for correctly discriminating a test photo's VCDR that was 25% smaller than the standard photo's VCDR. The VCDR uncertainty range for participants in Year 2, Year 3, and

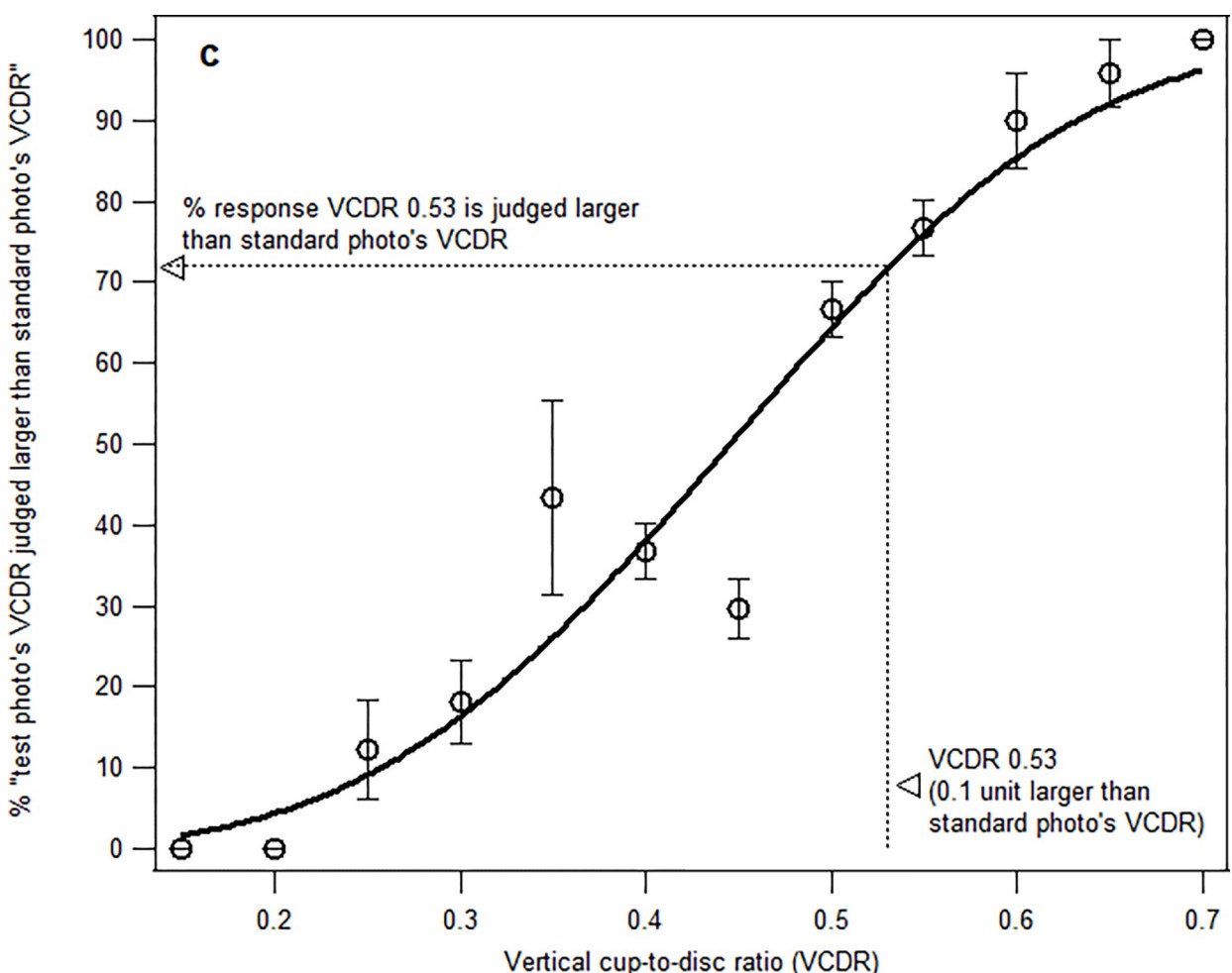

**Fig 4. Determination of % of times a VCDR of 0.53 judged to be larger than the standard photo's VCDR (0.43), that is, when the test photo VCDR was larger by 0.1 unit.**

Year 4 was 0.18±0.06, 0.17±0.05, and 0.18±0.03, respectively. There was no statistically significant difference in the VCDR uncertainty range between the three groups (one-way ANOVA [$F(2,27) = 0.121$, $p = 0.887$]).

The percentage of correctly estimating a test photo with VCDR of 0.53 indicated the ability of the participants to discriminate VCDR that was 0.10 units larger than the standard photo's VCDR (0.43). The mean percentage correct 0.53 VCDR estimation for participants in Year 2, 3, and 4 were 73.72%±11.61, 77.52%±14.23 and 79.56%±6.80, respectively. Although trainees with the most clinical experience (Year 4) tend to have higher percentages in judging the

**Table 1. Participants' sociodemographics and cumulative grade point average (CGPA).**

| Characteristics | Year 2 (n = 10) | Year 3 (n = 10) | Year 4 (n = 10) | Overall (n = 30) |
|---|---|---|---|---|
| Age | 21.30±0.68 | 22.00±0.00 | 23.20±0.42 | 22.17±0.91 |
| Gender | | | | |
| Male | 2 | 2 | 1 | 5 |
| Female | 8 | 8 | 9 | 25 |
| CGPA (out of 4) | 3.58±0.34 | 3.42±0.25 | 3.35±0.15 | 3.45±0.27 |

VCDR correctly, the differences in VCDR estimation between the participant groups were not statistically significant (one-way ANOVA [$F_{(2,27)} = 0.687$, $p = 0.512$]).

## Discussion

Despite technological advances in automated medical imaging, the competency in evaluating clinical findings by trainees in clinical courses such as optometry remains vital for the successful care of patients. This study assessed the ability to discriminate VCDR of the optic nerve head in clinical optometry trainees from three cohorts, each with different clinical experience and exposure. Generally, Year 2 participants had higher thresholds to achieve the point of subjective equality than Year 3 and Year 4 participants. Year 2 participants also had the lowest percentage correct to discriminate VCDR that was 0.10 larger than the median VCDR, compared to their Year 3 and Year 4 peers. However, all of these results were not statistically significant. In addition, the VCDR uncertainty range was also not significantly different between the cohorts. These results suggest that the assumed difference in the hands-on clinical training did not significantly influence the participants' ability to discriminate VCDR.

The COVID-19 pandemic resulted in the suspension of face-to-face teaching, including hands-on practical sessions. The current study's Year 2 participants spent significantly fewer hours in the teaching lab as most teachings were carried out online due to the COVID-19 lockdown. Despite spending less than a semester of hands-on clinical training, they were able to judge VCDR, based on fundus photos, just as well as their more senior peers. These findings suggest that online education and web-based teaching sessions, which replaced traditional teaching sessions, were effective, in line an earlier report for ophthalmology training [13]. Nevertheless, it can be argued that the lack of hands-on session could create a delay in the acquisition of necessary clinical skills such as ophthalmoscopy. It has been reported that that online tutorials were helpful in medical ophthalmic skill teaching [14]. However, these authors also reported that online teaching of direct ophthalmoscopy, one of the clinical procedures that allow VCDR evaluation, was challenging. As clinical competencies are found to be positively correlated with years of clinical experience [10, 15], the association between the ability to judge VCDR based only on fundus photos and the actual clinical competency to obtain it remains to be seen should be investigated in a future study.

This study's findings were in line with an earlier report where the clinical accuracy in evaluating the glaucomatous optic nerve head characteristics by first- and third-year ophthalmology residents were compared [16]. It was found that the third-year residents only did better than the first-year residents in evaluating four out of ten optic nerve head characteristics. These findings were attributed to theoretical teaching that was primarily conducted in the first year of the residents' training. Besides, another study reported no significant correlation between VCDR's graders' reliability and their years of experience in grading [6]. In our study, the clinical optometry trainees were taught about the ways to discriminate VCDR when they were in Year 2. In Years 3 and 4, more intensive practical training was conducted in order to improve clinical accuracy. Although the VCDR discrimination abilities of the current study's participants were not statistically different, their actual clinical ability to conduct fundus examination, such as direct ophthalmoscopy, may not be equal.

The results of this study appear to be in contrast to a study that found that medical students performed better VCDR evaluation than non-clinical graders [5]. However, their non-clinical graders had no prior experience with VCDR grading nor with clinical care in general. However, the current study's Year 2 participants did have some clinical training, albeit the least compared to their Year 3 and Year 4 peers. Therefore, it is reasonable to suggest that the experience that the Year 2 participants had contributed to their ability to discriminate VCDR.

For the current study's participants, at least, it seems that clinical experience was not the primary factor that influences VCDR discrimination. Participants in this study had a maximum of two-year gap in their clinical experience. A larger gap in clinical experience probably would lead to a different outcome. A future study could be conducted to compare the discrimination of VCDR between optometry trainees and registered optometrists with at least 5 to 10 years of experience. Moreover, the lack of hands-on experience during the COVID-19 pandemic may have contributed to differences in the ability to judge VCDR, even though the participants' clinical experiences were technically different. Across the three trainee groups, their ability to correctly discriminate 0.1 unit change in VCDR was similar at approximately 80%. Therefore, optimisation of the clinical ability may be achieved with additional lectures [17, 18] and clinical training [19, 20]. Indeed, the COVID-19 pandemic offers an opportunity for clinical teaching institutions to reshape medical training, with the use of new technology as part of educational tools [13]. Web-based teaching, clinical simulators, and remote mentoring offer the necessary flexibility to reverse any gaps in clinical skills due to the suspension of traditional teaching, while simultaneously be part of a future-ready curriculum.

This study has several limitations. One issue that arises is whether the judgements of the ONH and VCDR based on fundus photos, which lack depth cues, could accurately reflect the actual clinical ability of the trainees. However, some popular clinical methods of ONH observation also involve monocular or 2-dimensional observation, including the use of a monocular hand-held direct ophthalmoscope and evaluation of fundus photos captured with a fundus camera, which are relevant to the method employed in this study. Nevertheless, we also agree that there could be limitations on the transferability of the estimated 2D estimations of the VCDR, particularly when its examination involves binocular procedures such as the binocular indirect ophthalmoscopy and or with the Hruby lens. Secondly, although there were 99 test photos as stimuli, there were only a few fundus photos at the extreme ends of the VCDR values (n = 2 for 0.15 VCDR and n = 2 for 0.70 VCDR). Thirdly, the objective discrimination of VCDR was carried out by one person only (author LYR), which might have led to expectation bias.

## Conclusions

Optometry trainees with different years of clinical training experience have similar thresholds for discriminating VCDR when 2-dimensional fundus photos were used as the stimuli.

## Supporting information

**S1 Data.**
(XLSX)

## Acknowledgments

Special thanks to the participants who were involved in this study.

## Author Contributions

**Conceptualization:** Mohd Izzuddin Hairol.

**Data curation:** Mohd Izzuddin Hairol.

**Formal analysis:** Mohd Izzuddin Hairol, Yun Rou Lee.

**Investigation:** Mohd Izzuddin Hairol, Yun Rou Lee.

**Methodology:** Mohd Izzuddin Hairol.

**Project administration:** Mohd Izzuddin Hairol, Yun Rou Lee.

**Resources:** Mohd Izzuddin Hairol.

**Software:** Mohd Izzuddin Hairol.

**Supervision:** Mohd Izzuddin Hairol.

**Validation:** Mohd Izzuddin Hairol.

**Visualization:** Mohd Izzuddin Hairol.

**Writing – original draft:** Mohd Izzuddin Hairol, Yun Rou Lee.

**Writing – review & editing:** Mohd Izzuddin Hairol.

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
