## [Decision Letter · Decision Letter 0]

6 Jul 2022

PONE-D-22-10365Accuracy of vertical cup-to-disc ratio discrimination among clinical optometry trainees with different years of clinical experiencePLOS ONE

Dear Dr. Mohd Izzuddin Hairol,

Thank you for submitting your manuscript to PLOS ONE. After careful consideration, we feel that it has merit but does not fully meet PLOS ONE’s publication criteria as it currently stands. Therefore, we invite you to submit a revised version of the manuscript that addresses the points raised during the review process.

We look forward to receiving your revised manuscript.

Kind regards,

Alon Harris

Academic Editor

PLOS ONE

Journal Requirements:

Reviewers' comments:

Reviewer's Responses to Questions

**Comments to the Author**

1. Is the manuscript technically sound, and do the data support the conclusions?

Reviewer #1: Partly

Reviewer #2: Yes

2. Has the statistical analysis been performed appropriately and rigorously? 

Reviewer #1: Yes

Reviewer #2: Yes

3. Have the authors made all data underlying the findings in their manuscript fully available?

Reviewer #1: Yes

Reviewer #2: Yes

4. Is the manuscript presented in an intelligible fashion and written in standard English?

Reviewer #1: Yes

Reviewer #2: Yes

5. Review Comments to the Author

Reviewer #1: The main question that comes forward is qualifying as a technically sound manuscript. Following the description above "The manuscript must describe a technically sound piece of scientific research with data that supports the conclusions. Experiments must have been conducted rigorously, with appropriate controls, replication, and sample sizes. The conclusions must be drawn appropriately based on the data presented." I am concerned that the research does not necessarily support the final conclusion statement as written, or at least a better description of how the conclusion is a leap from the data. It is well know that fundus photography does not show depth perception, thus to conclude that optometry trainees do or do not discriminate appropriately the VCDR off only 2 dimension photography seems to be a bit of scientific stretch. The conclusion only applies to whether optometry trainees discriminate the VCDR well in fundus photos, with lack of depth clues, not their performance clinically. There seems to be a leap of faith that the performance on the grading of photography (research data) will be the same as the trainees performance in clinic, per the conclusion. I encourage the authors to include a discussion on the limitations and transferability of measurements estimated 2 dimensional photography versus 3 dimensional clinically.

The figures and tables were accurately reported. Statistics seemed appropriate and accurate. There was a case for using an n=30 but study was conducted on a much larger number and should only make statistics stronger (line 70-76) if including additional subjects. Methods appeared to be sound (line 148-150). There were a few discussion points regarding COVID and training (line 199) that start to delve into some explanation behind the results, however, I think time spent on the conclusion on above relevant topics would be prudent and encourage authors to include further detail. Strong references (line 34, 204) were including. Overall, a interesting study that may further help guide Optometry curriculum.

Reviewer #2: The authors presented an interesting study evaluating the accuracy of vertical cup-to-disc ratio discrimination among clinical optometry trainees with different years of clinical experience. The manuscript is clear, its topic is original in content, and the conclusions are consistent with the evidence presented. The manuscript is with merit and the findings are worth reporting, but the authors should address the following comments before publication.

- Introduction/entire manuscript

o It would be helpful to have a graphical visualization of the optic nerve head and of the vertical cup to disc ratio for readers that are not experts in ophthalmology

o Revise the use of abbreviations: an abbreviation should be explained once in the manuscript and after only the abbreviation (and not the full explanation) should be used – revise the manuscript correspondingly (i.e. cup-to-disc ratio (CDR), horizontal cup to disc ratio (HCDR) and vertical cup to disc ratio (VCDR) are explained in the introduction: only the abbreviations should be used after in the entire manuscript – for example at line 84 “Every fundus photo's vertical cup-to-disc ratio (VCDR)” should be replaced by “Every fundus photo's VCDR” )

- Methodology

o Section “study participants”

The authors should provide the details of the name and location of the University/optometrist clinic where the study was conducted

The authors should provide a brief explanation of what the “cumulative grade point average (CGPA)” is, for readers who may not be familiar with such score

o Section “Experimental Stimuli and Instrument”: how was the selection of the one hundred fundus photos made? One or multiple operators and at which level of training? The authors should provide additional details in this section

- Discussion

o The authors should provide insight about the future directions of this research and discuss the significance (from a clinical/educational point of view) of their results

- Figures and Tables legends: the authors should provide the explanations of the abbreviations used in the legends

6. PLOS authors have the option to publish the peer review history of their article (what does this mean?). If published, this will include your full peer review and any attached files.

Reviewer #1: No

Reviewer #2: No

---

## [Author Response · Author response to Decision Letter 0]

11 Jul 2022

Manuscript ID: 

PONE-D-22-10365

Manuscript Title: 

Accuracy of vertical cup-to-disc ratio discrimination among clinical optometry trainees with different years of clinical experience

Response to Reviewers 

We thank both Reviewers for their helpful comments and suggestions. They have helped us significantly improved the manuscript. We address their comments point-by-point below.

Reviewer #1: 

Reviewer #1 found that the study was interesting that may further guide the Optometry curriculum. They also found that our figures, tables, statistics, and methods sound and accurately reported. We agreed that our initial conclusion might have appeared like a leap between the ability to discriminate VCDR to clinical competency; however, we have addressed this concern by making the necessary changes as described in detail below. 

Comment 1: 

I encourage the authors to include a discussion on the limitations and transferability of measurements estimated 2 dimensional photography versus 3 dimensional clinically.

Our response:

We thank Reviewer 1 for their helpful comments, particularly when the study was found to be interesting and useful to further guide the relevant optometry curriculum. 

We agree with Reviewer 1’s comment on the transferability of estimated measurements of VCDR based on 2-D photos vs. the 3D nature of the optic nerve head. Nevertheless, even clinically, some popular methods of ONH observation also involve 2D observation. These include the use of a monocular hand-held direct ophthalmoscope and evaluation of the fundus photos captured with a fundus camera, such as the one used in this study. The fundus-matching method proposed by Kwok et al 2017 , where ophthalmology residents were tasked to match real patients’ actual fundus with their corresponding 2D fundus photos, has been found to be effective in improving clinical judgement and skills. 

Having said that, we also agree that there could be limitations on the transferability of the estimated 2D estimations of the ONH, particularly when its examination involves binocular procedures such as the binocular direct ophthalmoscopy and or using the Hruby lens. We address this as limitations in the revised manuscript, lines 255-263: 

“One issue that arises is whether the judgements of the ONH and VCDR based on fundus photos, which lack depth cues, could accurately reflect the actual clinical ability of the trainees. However, some popular clinical methods of ONH observation also involve monocular or 2-dimensional observation, including the use of a monocular hand-held direct ophthalmoscope and evaluation of fundus photos captured with a fundus camera, which are relevant to the method employed in this study. Nevertheless, we also agree that there could be limitations on the transferability of the estimated 2D estimations of the VCDR, particularly when its examination involves binocular procedures such as the binocular indirect ophthalmoscopy and or with the Hruby lens.” 

We have also revised the Conclusion: 

(i) of the abstract (lines 17-19)

“Although Year 4 optometry trainees performed slightly better at estimating VCDR than their Year 3 and Year 2 peers, the number of years of clinical experiences did not significantly affect their VCDR discrimination thresholds when 2-dimensional fundus photos were used as stimuli.” 

(ii) and in Conclusions (lines 266-267)

“Optometry trainees with different years of clinical training experience have similar thresholds for discriminating VCDR when 2-dimensional fundus photos were used as the stimuli.” 

Comment 2:

There was a case for using an n=30 but study was conducted on a much larger number and should only make statistics stronger (line 70-76) if including additional subjects.

Our response:

As mentioned in the subheading Participant Sampling, the n=30 of the current study has adequate power (80%). Although n could have been made larger, the discrimination thresholds measured were based on a large number of observations, where each participant completed 297 trials (99 trials × 3 runs), excluding training sessions. Such a paradigm is said to be optimum for identifying systematic functional relationships in experimental designs with (relatively) small n (Smith & Little, 2018) . Thus, for the purpose of the study, n=30 is adequate although we do not disagree with Reviewer 1 that a larger sample size should make the statistics stronger. 

Comment 3: 

There were a few discussion points regarding COVID and training (line 199) that start to delve into some explanation behind the results, however, I think time spent on the conclusion on above relevant topics would be prudent and encourage authors to include further detail.

Our response: 

Yes we agree that the impact of the pandemic on training is prudent to be discuss in detail. We have revised and added the point below in Discussion, lines 204-218:

“The COVID-19 pandemic resulted in the suspension of face-to-face teaching, including hands-on practical sessions. The current study’s Year 2 participants spent significantly fewer hours in the practical teaching lab as most teachings were carried out online due to the COVID-19 lockdown. Despite spending less than a semester of hands-on clinical training, they were able to judge VCDR based on fundus photos just as well as their more senior peers. These findings suggest that online education and web-based teaching sessions, which replaced traditional teaching sessions, were effective, in line with an earlier report for ophthalmology training (Ferrera et al., 2020). Nevertheless, it can also be argued that the lack of hands-on session could create a delay in the acquisition of necessary clinical skills such as ophthalmoscopy. It has been reported that online tutorials were helpful in medical ophthalmic skill teaching (Shih et al., 2020). However, these authors also reported that online teaching of direct ophthalmoscopy, one of the clinical procedures that allow VCDR evaluation, was challenging. As clinical competencies are found to be positively correlated with years of clinical experience (Barsuk et al., 2017; Kong et al., 2011), the association between the ability to judge VCDR based only on fundus photos and actual clinical competency to obtain it remains to be seen and should be investigated in a future study.”

In the same vein, we have also include the point on how clinical training can be enhanced, as part of the future direction after the COVID-19 pandemic, lines 249-254: 

“Indeed, the COVID-19 pandemic offers an opportunity for clinical teaching institutions to reshape medical training, with the use of new technology as part of educational tools (Ferrera et al., 2020). Web-based teaching, clinical simulators, and remote mentoring offer the necessary flexibility to reverse any gaps in clinical skills due to the suspension of traditional teaching while simultaneously be part of a future-ready curriculum.” 

Reviewer #2

We would also like to thank Reviewer2 for their valuable comments. They found that the manuscript to be with merit, and our findings worth reporting. The comments are addressed in detail below. 

Comment 1:

It would be helpful to have a graphical visualization of the optic nerve head and of the vertical cup to disc ratio for readers that are not experts in ophthalmology

Our response:

We thank Reviewer2 for the suggestion. Indeed, a figure of the optic nerve head and of the VCDR calculation would be helpful. We have addressed this comment in the Introduction as Figure 1. As such, the numbering of all other figures has been modified accordingly. 

Comment 2: 

Revise the use of abbreviations: an abbreviation should be explained once in the manuscript and after only the abbreviation (and not the full explanation) should be used – revise the manuscript correspondingly (i.e. cup-to-disc ratio (CDR), horizontal cup to disc ratio (HCDR) and vertical cup to disc ratio (VCDR) are explained in the introduction: only the abbreviations should be used after in the entire manuscript – for example at line 84 “Every fundus photo's vertical cup-to-disc ratio (VCDR)” should be replaced by “Every fundus photo's VCDR” )

Our response:

We thank Reviewer2 for the suggestion. The manuscript has been revised where all abbreviations are used appropriately as suggested. These changes are tracked in the manuscript. 

Comment 3:

The authors should provide the details of the name and location of the University/optometrist clinic where the study was conducted

Our response:

The name and location of the university where the study was conducted are now stated in the revised manuscript. The first line (line 62) of the Study Participants section now reads: 

‘This cross-sectional study was carried out from February to July 2021 in the Optometry Clinic, Universiti Kebangsaan Malaysia (the National University of Malaysia) located in Kuala Lumpur.’

Comment 4:

The authors should provide a brief explanation of what the “cumulative grade point average (CGPA)” is, for readers who may not be familiar with such score. 

Our response:

We have provided a brief explanation on the CGPA to aid understanding for readers who may not be familiar with the score. The Study Participants section has now been revised to address this point (lines 68-71):

‘All participants had a cumulative grade point average (CGPA) of at least 3.00 out of a maximum of 4.00. It is calculated as the mean of the total grade points for every semester divided by the total number of credits where ≥3.00 CGPA reflected an overall academic performance graded as ‘distinction’.’ 

Comment 5:

Section “Experimental Stimuli and Instrument”: how was the selection of the one hundred fundus photos made? One or multiple operators and at which level of training? The authors should provide additional details in this section

Our response:

The details on the selection of the fundus photos have been revised and are now described in Experimental Stimuli and Instrument as follows (lines 87-90):

‘The fundus photos were first compiled from the instrument’s database by author YRL, a final-year optometry student. They were then evaluated and selected by consensus by YRL, author MIH who was a registered optometrist and academician with a 15-year of experience, and another optometrist who was independent of the study.’ 

Comment 6:

The authors should provide insight about the future directions of this research and discuss the significance (from a clinical/educational point of view) of their results

Our response:

We have also added that the integration of new technology as an important tool to reverse any gaps in clinical skills, and also as part of a future ready curriculum in medical training (Discussion lines 249-254): 

“Indeed, the COVID-19 pandemic offers an opportunity for clinical teaching institutions to reshape medical training, with the use of new technology as part of educational tools (Ferrera et al., 2020). Web-based teaching, clinical simulators, and remote mentoring offer the necessary flexibility to reverse any gaps in clinical skills due to the suspension of traditional teaching while simultaneously be part of a future-ready curriculum.” 

Comment 7:

- Figures and Tables legends: the authors should provide the explanations of the abbreviations used in the legends

Our response:

The abbreviation used in the Figures (i.e. VCDR) is provided in the axis labels. Abbreviations are also now mentioned in full for Table 1. 

END OF RESPONSE 

Mohd Izzuddin Hairol

9 July 2022

---

## [Decision Letter · Decision Letter 1]

5 Sep 2022

Accuracy of vertical cup-to-disc ratio discrimination among clinical optometry trainees with different years of clinical experience

PONE-D-22-10365R1

Dear Dr. Mohd Izzudin Hairol,

We’re pleased to inform you that your manuscript has been judged scientifically suitable for publication and will be formally accepted for publication once it meets all outstanding technical requirements.

Kind regards,

Alon Harris

Academic Editor

PLOS ONE

Additional Editor Comments (optional):

Reviewers' comments:

Reviewer's Responses to Questions

**Comments to the Author**

1. If the authors have adequately addressed your comments raised in a previous round of review and you feel that this manuscript is now acceptable for publication, you may indicate that here to bypass the “Comments to the Author” section, enter your conflict of interest statement in the “Confidential to Editor” section, and submit your "Accept" recommendation.

Reviewer #2: All comments have been addressed

Reviewer #3: All comments have been addressed

2. Is the manuscript technically sound, and do the data support the conclusions?

Reviewer #2: Yes

Reviewer #3: Yes

3. Has the statistical analysis been performed appropriately and rigorously? 

Reviewer #2: Yes

Reviewer #3: Yes

4. Have the authors made all data underlying the findings in their manuscript fully available?

Reviewer #2: Yes

Reviewer #3: Yes

5. Is the manuscript presented in an intelligible fashion and written in standard English?

Reviewer #2: Yes

Reviewer #3: Yes

6. Review Comments to the Author

Reviewer #2: The authors addressed all my comments to the best of their knowledge and the manuscript can be accepted for publication.

Reviewer #3: The authors addressed all the comments of the reviewers, and corrected and added accordingly the manuscript

7. PLOS authors have the option to publish the peer review history of their article (what does this mean?). If published, this will include your full peer review and any attached files.

Reviewer #2: No

Reviewer #3: No

---

## [Editor Report · Acceptance letter]

8 Sep 2022

PONE-D-22-10365R1 

Accuracy of vertical cup-to-disc ratio discrimination among clinical optometry trainees with different years of clinical experience 

Dear Dr. Hairol:

I'm pleased to inform you that your manuscript has been deemed suitable for publication in PLOS ONE. Congratulations! Your manuscript is now with our production department. 

Kind regards, 

on behalf of

Dr. Alon Harris 

Academic Editor

PLOS ONE